# Detection and analysis of Serpin and RP26 specific antibodies for monitoring *Schistosoma haematobium* transmission

**Mio Kokubo-Tanaka**[1,2,3], **Anna Overgaard Kildemoes**[4], **Evans Asena Chadeka**[1,2,5], **Benard Ngetich Cheruiyot**[1,5], **Taeko Moriyasu**[1,2,6,7], **Miho Sassa**[1], **Risa Nakamura**[1,2,7], **Mihoko Kikuchi**[7,8], **Yoshito Fujii**[9], **Claudia J. de Dood**[10], **Paul L. A. M. Corstjens**[10], **Satoshi Kaneko**[2,5,6], **Haruhiko Maruyama**[3], **Sammy M. Njenga**[1,11], **Remco de Vrueh**[12], **Cornelis Hendrik Hokke**[4], **Shinjiro Hamano**[1,2,5,7] *

1 Department of Parasitology, Institute of Tropical Medicine (NEKKEN), Nagasaki University, Nagasaki, Japan, 2 Program for Nurturing Global Leaders in Tropical and Emerging Communicable Diseases, Graduate School of Biomedical Sciences, Nagasaki University, Nagasaki, Japan, 3 Division of Parasitology, Department of Infectious Diseases, Faculty of Medicine, University of Miyazaki, Miyazaki, Japan, 4 Department of Parasitology, Leiden University Center of Infectious Diseases (LUCID), Leiden University Medical Center, Leiden, The Netherlands, 5 Nagasaki University, Kenya Research Station, NUITM-KEMRI Project, Nairobi, Kenya, 6 Department of Eco-Epidemiology, Institute of Tropical Medicine (NEKKEN), Nagasaki University, Nagasaki, Japan, 7 The Joint Usage/Research Center on Tropical Disease, Institute of Tropical Medicine (NEKKEN), Nagasaki University, Nagasaki, Japan, 8 Department of Immunogenetics, Institute of Tropical Medicine (NEKKEN), Nagasaki University, Nagasaki, Japan, 9 Department of Medical Technology, Sanyo Women's College, Hatsukaichi, Japan, 10 Department of Cell and Chemical Biology, Leiden University Medical Center, Leiden, The Netherlands, 11 Eastern and Southern Africa Centre of International Parasite Control (ESACIPAC), Kenya Medical Research Institute (KEMRI), Nairobi, Kenya, 12 Lygature, Utrecht, The Netherlands

* shinjiro@nagasaki-u.ac.jp

**Data Availability Statement:** All relevant data are within the manuscript and its Supporting information files.

## Abstract

### Background

*Schistosoma haematobium* is the causative pathogen for urogenital schistosomiasis. To achieve progress towards schistosomiasis elimination, there is a critical need for developing highly sensitive and specific tools to monitor transmission in near-elimination settings. Although antibody detection is a promising approach, it is usually unable to discriminate active infections from past ones. Moreover, crude antigens such as soluble egg antigen (SEA) show cross-reactivity with other parasitic infections, and it is difficult to formulate the standard preparations. To resolve these issues, the performances of recombinant antigens have been evaluated. The antibody responses against recombinant *S. haematobium* serine-protease inhibitor (ShSerpin) and RP26 were previously shown to reflect active schistosome infection in humans. Furthermore, antibody detection using multiple recombinant antigens has been reported to improve the accuracy of antibody-based assays compared to single-target assays. Therefore, we examined the performances of ShSerpin, RP26 and the mixture of these antigens for detecting *S. haematobium* low-intensity infection and assessed the potential for transmission monitoring.

**Funding:** The Global Health Innovative Technology Fund (T2017-272), a Grants-in-Aid for International Scientific Research (A) by JSPS (17H01684 and 21H04852), and the International Collaborative Research Program: Science and Technology Research Partnership for Sustainable Development (SATREPS) by JICA and AMED (JP23jm0110027) to SH. The funders had no role in study design, data collection and analysis, decision to publish, or preparation of the manuscript.

**Competing interests:** The authors have declared that no competing interests exist.

## Methodology/Principal findings

We collected urine and plasma samples from school-aged children in Kwale, Kenya and evaluated *S. haematobium* prevalence by number of eggs in urine and worm-derived circulating anodic antigen (CAA) in plasma. Among 269 pupils, 50.2% were CAA-positive by the lateral flow test utilizing up-converting phosphor particles (UCP-LF CAA), while only 14.1% were egg-positive. IgG levels to *S. haematobium* SEA (ShSEA), ShSerpin, RP26, and the mixture of ShSerpin and RP26 were measured by ELISA. The mixture of ShSerpin and RP26 showed the highest sensitivity, 88.7%(125/141)among the four antigens in considering indecisive UCP-LF CAA results as negative.

## Conclusion/Significance

IgG detection against the ShSerpin-RP26 mixture demonstrated better sensitivity for detection of active *S. haematobium* infection. This recombinant antigen mixture is simpler to produce with higher reproducibility and can potentially replace ShSEA in monitoring transmission under near-elimination settings.

## Author summary

Schistosomiasis is a parasitic disease affecting more than 200 million people. Currently, ongoing schistosomiasis control programs are based on mass drug administration with declining prevalence in many regions. Under near-elimination settings, highly sensitive, specific and feasible tools for monitoring transmission are essential. Among the existing standard methods, antibody detection against soluble egg antigen (SEA) is the most sensitive tool for detecting low-intensity infection. However, the specificity is generally low due to the longevity of antibody responses after treatment and SEA's inherent cross-reactive potential. To overcome the drawbacks of conventional crude antigens in antibody detection assays, we used a combination of single recombinant antigens, ShSerpin and RP26. Here, we show that antibodies to this antigen mix indicate *S. haematobium* infection with high sensitivity and specificity. Notably, cross-reactivities to other helminthic infections are markedly reduced compared to ShSEA, and antigen-specific IgG levels correlate positively with circulating anodic antigen (CAA) concentration that reflects active infection intensity. Our results suggest that the mixture of ShSerpin and RP26 is superior to ShSEA for IgG based detection of active *S. haematobium* infection. The excellent performance of this antigen mixture shows promise for applications in schistosomiasis monitoring under near-elimination settings.

## Introduction

Schistosomiasis is a water-borne parasitic infection caused by trematodes of the genus *Schistosoma* and is a neglected tropical disease (NTD) affecting more than 200 million people worldwide, especially in sub-Saharan Africa [1]. Among the five species that cause human schistosomiasis, *Schistosoma haematobium* is responsible for urogenital schistosomiasis and causes severe morbidities such as obstructive uropathy and bladder cancer [2]. It also causes genital schistosomiasis, affecting reproductive health [3]. In endemic regions, school-aged children are the most vulnerable group [2], and they are the main target of deworming programs

based on mass drug administration (MDA) [4], which lowers prevalence and infection intensity [5,6]. In the 2021–2030 NTD Roadmap [7], WHO targets eliminating schistosomiasis through intensified deworming interventions such as MDA, in which highly sensitive diagnostics are essential to monitor transmission during and after the interventions and assess the outcome of the control programs [8].

In *S. haematobium* endemic areas, infection is commonly monitored by detecting eggs in urine by microscopy, which is highly specific but lacks sensitivity to detect low-intensity infections [2,9,10]. Therefore, *S. haematobium* prevalence is frequently underestimated in settings with low endemicity. On the other hand, detecting schistosome circulating anodic antigen (CAA) in serum or urine by a lateral flow test utilizing quantitative up-converting reporter particles (UCP-LF CAA assay) is much more sensitive in detecting low-intensity infection [11–13]. CAA becomes detectable several weeks after infection [14], reflects the worm burden, and is cleared after successful treatment [15]. Therefore, CAA detection using the UCP-LF CAA assay is now one of the most sensitive and specific methods [10,16,17] and it is an excellent way to identify active infection. Currently, UCP-LF CAA is a laboratory-based test requiring the pre-treatment of samples and which has been implemented in the routine examination at LUMC (Leiden University Medical Center) and at AMPATH (The Academic Model Providing Access to Healthcare, South Africa). However, it is not commercially available [18].

A sensitive method of monitoring infection and/or transmission in low-endemic settings is antibody detection [13,19,20]. Enzyme-linked immunosorbent assay (ELISA) using crude antigens is a standard method. In particular, soluble egg antigen (SEA), a preparation consisting of hundreds of schistosome egg proteins and glycoproteins, is widely used as coating mixture [21], and ELISA kits are commercially available [20]. Despite its high sensitivity, the fundamental disadvantage of antibody detection to SEA is the longevity of antibodies. As a result, it cannot fully distinguish active infections from previous, cleared infections [13]. The antibody level to SEA also cannot be used to indicate infection intensity, which is an important parameter in assessing morbidity [22]. Furthermore, crude antigens comprise numerous cross-reactive antigenic molecules, posing limitations to assay specificity [23,24].

Various recombinant antigens have been tested to overcome the drawbacks of antibody detection using crude antigens [25]. Previously, IgG detection against recombinant serine-protease inhibitor (Serpin) and recombinant Sm22.3 (RP26) were reported to reflect active schistosome infection in humans [26]. Serpin is expressed in stages parasitic to humans [27,28]. Detection of IgG to recombinant *S. haematobium* Serpin (ShSerpin) is sensitive and specific, with a species preference trend [26,29]. On the other hand, Sm22.3 from *S. mansoni*, is expressed in cercariae, schistosomula, and immature and adult worms but not in eggs [30,31]. Sequences with high homology to Sm22.3 are also found in unnamed protein products of *S. haematobium*, and elevated IgG anti-recombinant Sm22.3 (RP26) levels have been found to be associated with both *S. mansoni* or *S. haematobium* infections [26,32]. Single recombinant antigens sometimes show insufficient sensitivity compared to crude antigens, but the diagnostic performances are improved when combined in a cocktail with one or more other antigens [33–36]. Therefore, it is hypothesized that antibody detection using a mix of ShSerpin and RP26 would be useful as a monitoring tool in the process towards schistosomiasis haematobia elimination.

In the current study, we assessed the performance of antibody detection against ShSerpin, RP26 and the mixture of the two antigens in plasma samples collected in Kwale, *S. haematobium* endemic area in Kenya. Based on a prior study, we enrolled children in schools with low infection intensity to verify the ability of these antigens to detect low-intensity infections in a serological assay [37]. IgG to the mixture of ShSerpin and RP26 detected active *S. haematobium* infection with high sensitivity. Therefore, we propose the ShSerpin/RP26 IgG assay as a promising approach for monitoring transmission in near elimination settings.

## Methods

### Ethical considerations

This study was reviewed and approved by the scientific and ethics review unit of Kenya Medical Research Institute (KEMRI) (SSC No. 2084) and the ethical review board of the Institute of Tropical Medicine, Nagasaki University (NUITM) (No. 140829127–7). Before the field activities in primary schools in Kwale, we held meetings with parents/guardians, school administrators and teachers to explain the objective and the study methods. After children assented, written consent forms were obtained from their parents/guardians, and then blood, urine and stool samples were collected from the consented children. After the analyses, all the children confirmed to be infected with schistosomes either by urine egg or plasma CAA detection were treated with 40 mg/kg praziquantel (Prazitel, Cosmos Ltd., Nairobi, Kenya). For those who were found to be infected with soil-transmitted helminths (STH) by the Kato-Katz technique, treatments with 400 mg albendazole (ABZ Tablet, Indoco Remedies Ltd., Mumbai, India) were given. The drugs were administered according to WHO guidelines [4], and the children were offered a light meal to eat before swallowing tablets. The local medical officers confirmed that every child swallowed the tablets and further observed them to manage any adverse effects.

For negative controls, blood sample collection from healthy Japanese volunteers was approved by the NUITM ethical review board (No. 140829127–7), and written consent forms were obtained from the volunteers. For cross-reactivity evaluations, we used serum/plasma samples that were positive for other helminth infections. The *S. mansoni* infection-positive samples were collected in Mbita, Kenya (the study approved by SSC No. 2084 in KEMRI and No. 140829127–7 in NUITM). Other samples (*S. japonicum* and other parasitic helminth infection-positive cases) were bio-banked clinical serum specimens stored in the Division of Parasitology, Department of Infectious Diseases, Faculty of Medicine, University of Miyazaki. The utilization of the residual samples was approved by the Research Ethics Review Board of the Faculty of Medicine, University of Miyazaki (No. O-0359).

### Study area and participants

We conducted a cross-sectional study in three primary schools in Kwale, Kwale County, located on the Kenyan south coast. Kwale is known to be endemic for *S. haematobium*. Three schools were selected for the present study, and the prevalence of *S. haematobium* infection by urine microscopy in the three schools in 2012 was 11.6%, 35.6%, and 33.8%, respectively. The mean infection intensities among the infection-positive pupils in these schools were 1–3 eggs/10 mL urine at that time [37]. We recruited 400 pupils from grades 1 to 7 in the schools. 335 pupils were consented, and 269 of them were finally included in the analysis based on the inclusion and exclusion criteria (**Fig 1**). The male/female ratio of these 269 was 1:1.3 (118 males and 151 females). The median of the school grades was 4, while we could not access the age data of the participants. We carried out the field survey from 10th to 14th June 2019, and the latest annual MDA was implemented in July 2018 in Kwale, 11 months before our sampling.

### Sample collection in Kwale

Experienced field surveyors and local medical staffs managed sample collection. Plasma, urine, and stool samples were collected from the participants. Each pupil was given a unique identification (ID) number for anonymization. Before the sample collection, the field surveyors delivered containers for stool samples to the schools, and school health teachers assisted pupils in

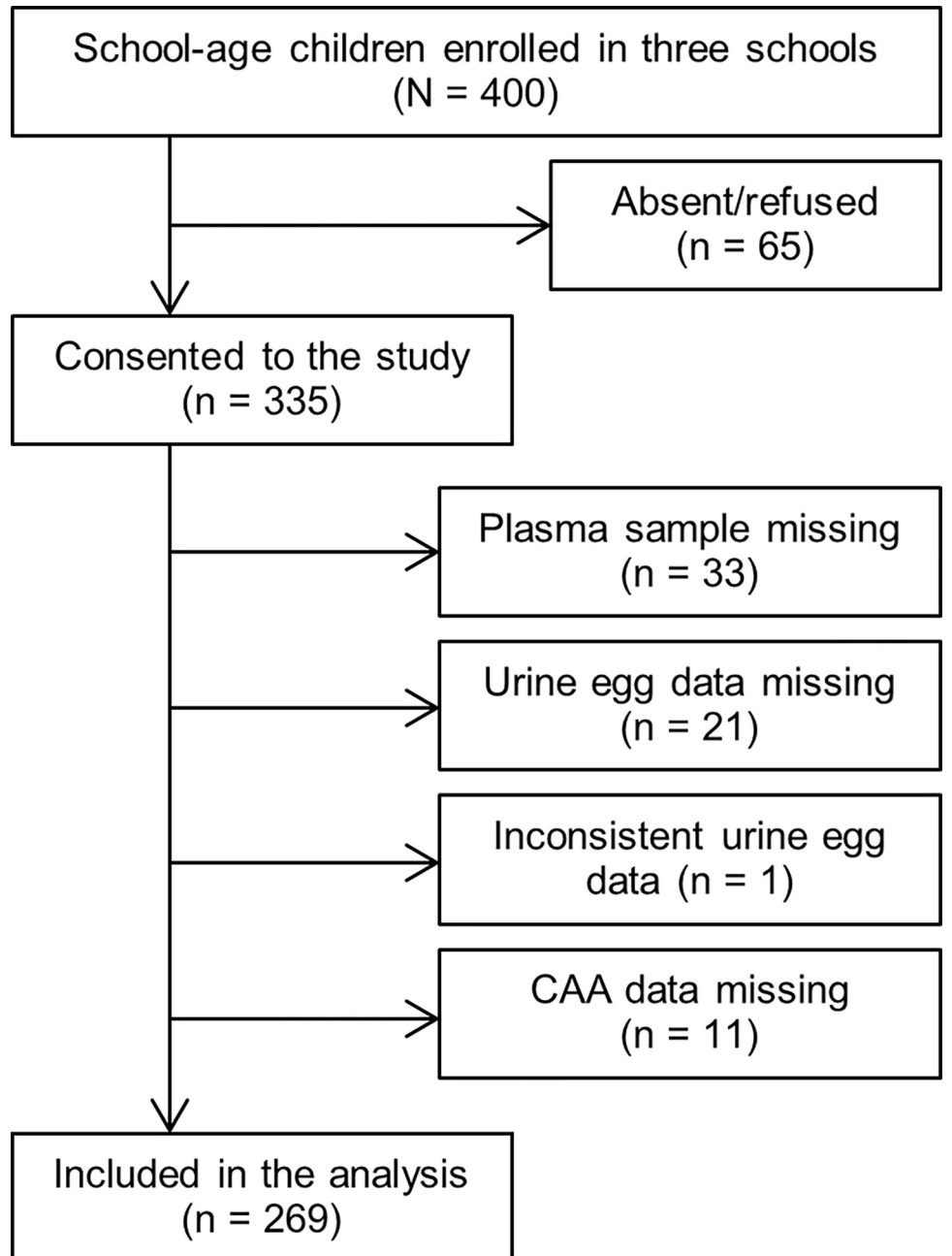

**Fig 1. Inclusion and exclusion criteria of the study participants.** Of the 400 pupils enrolled in the three schools, 269 were included in the analysis.

collecting the samples. The field surveyors and the local medical staffs visited the schools on the morning of the sample collection day. Blood samples were collected in sodium heparin tubes by experienced local medical staffs. Morning urine samples were collected in the container labelled with IDs. Early morning stool samples were also collected in the ID-labelled containers distributed in advance for two consecutive days. Then, all samples were transferred to the laboratory at the NUITM-KEMRI research station in Kwale. The blood samples were centrifuged, and 2 mL plasma was aliquoted for antibody detection by ELISA and CAA

detection by UCP-LF CAA assay. The plasma samples were stored in a -30˚C freezer in the KEMRI-NUITM research station laboratory. The samples were then transported to the Department of Parasitology, NUITM for ELISA and to LUMC for UCP-LF CAA assay.

### Egg detection in urine by microscopy

Morning urine samples were examined for *S. haematobium* egg using a standard urine filtration technique [9]. Briefly, 10 mL of urine was aliquoted using a syringe and filtered through a 12 μm polycarbonate filter membrane. The filtrates were examined under a microscope. Every sample was examined by two experienced laboratory technicians, and the results were expressed as the number of eggs per 10 mL of urine. The infection intensity of *S. haematobium* was categorized as light (1–49 eggs/10 mL urine) or heavy ($\geq$ 50 eggs/10 mL urine) based on the WHO guidelines [38]. The filter membranes were sent to the Department of Parasitology, NUITM, where the results were double-checked. When the results were inconsistent between the two laboratories, the sample was excluded from the analysis (n = 1, **Fig 1**).

### Circulating Anodic Antigen (CAA) detection in plasma samples

CAA concentrations were determined by UCP-LF assay using 500 μL trichloroacetic acid (TCA) precipitated plasma (PCAA500) as previously described [16,18], with 'P' indicating plasma and '500' indicating the approximate amount of plasma evaluated on the LF strip. The cut-off threshold of the assay was 1 pg/mL; samples with CAA concentrations above 1 pg/mL were regarded as positive, below 0.5 pg/mL as negative, and between 0.5 and 1 pg/mL as "indecisive". A relevant number of the indecisive samples may be positive when tested with higher volume. Most of this was published with urine samples [17,39] (**Table 1**). However, it is difficult to use relevantly larger volumes of plasma (and serum) due to increased viscosity of TCA-treated serum/plasma when using more than 500 μL. Therefore, PCAA500 positive results were further classified into three categories of infection-intensity based on the CAA concentration as follows: high, PCAA500 $\geq$ 100 pg/mL; medium, 10 $\leq$ PCAA500 < 100 pg/mL; low, 1 $\leq$ PCAA500 < 10 pg/mL; indecisive, 0.5 $\leq$ PCAA500 < 1 pg/mL.

**Table 1. *S. haematobium* infection prevalence and intensity determined by the results of pCAA500 and egg detection among 269 pupils.**

| | | | Egg detection in urine | | | Total n (%) | |
| --- | --- | --- | --- | --- | --- | --- | --- |
| | | | Positive | | Negative | | |
| | | | Heavy* | Light† | | | |
| PCAA500 | Positive‡ | High§ | 5 | 12 | 25 | 42 (15.6) | 135 (50.2) |
| | | Medium‖ | 1 | 11 | 29 | 41 (15.2) | |
| | | Low¶ | 0 | 3 | 49 | 52 (19.3) | |
| | Indecisive** | | 0 | 1 | 19 | | 20 (7.4) |
| | Negative†† | | 0 | 5 | 109 | | 114 (42.4) |
| | Total n (%) | | 6 (2.2) | 32 (11.9) | 231 (85.9) | | 269 |
| | | | | 38 (14.1) | | | |

The prevalence and the intensity of *S. haematobium* infection was determined based on the results of egg detection in 10 mL urine under microscopy and CAA detection in 500 μL plasma by UCP-LF CAA assay (PCAA500). Among the egg positive population, the geometric mean intensity was 9 (eggs/10 mL urine). Most of the egg positive population showed light infection.

*Heavy: $\geq$ 50 eggs/10 mL urine; †Light: 1–49 eggs/10 mL urine.

‡PCAA500 Positive: 1 pg/mL $\leq$ PCAA500; §High: PCAA500 $\geq$ 100 pg/mL; ‖Medium: 10 $\leq$ PCAA500 < 100 pg/mL; ¶Low:1 $\leq$ PCAA500 < 10 pg/mL; **Indecisive: 0.5 $\leq$ PCAA500 < 1 pg/mL; ††Negative: 0 $\leq$ PCAA500 < 0.5 pg/mL.

## Egg detection in stool by microscopy

*S. mansoni* and STH infection burden were assessed by the Kato-Katz method using 41.7 mg stool templates [40]. Two thick smears were prepared from single stool samples on two consecutive days. The smear slides were microscopically examined by two experienced laboratory technicians independently. The slides were examined within one hour for hookworm eggs and later for *S. mansoni*, *Trichuris trichiura*, and *Ascaris lumbricoides*. No diagnostic methods for *Strongyloides* were used. The technicians recorded egg counts per slide, and the results were later converted into eggs per gram of stool (EPGs) by multiplying a factor of 24. The geometric mean was determined based on EPGs for all slides.

## Estimation of the active schistosome infection

As there is no confirmatory diagnostic for active schistosome infection, we used the composite results of egg detection and PCAA500 to define the active infection status and to evaluate the diagnostic accuracy of antibody detection. A participant was considered positive for active *S. haematobium* infection (Sh+) if the urine egg or the PCAA500 were positive. A participant was considered negative (Sh-) if both results were negative.

## Antigen preparations

Crude *S. haematobium* soluble egg antigen (ShSEA) and recombinant antigens, ShSerpin (accession no. AAA19730) and Sm22.3 (accession no. AAB81008) were prepared for ELISA (described below). ShSEA used was produced from *Schistosoma haematobium* eggs isolated from infected Golden Syrian hamster livers. The liver material is from the *S. haematobium* life cycle upkept at LUMC since 1993 with a NAMRU-3, Cairo, Egypt strain. ShSEA was prepared on ice by simple mechanical disruption of *S. haematobium* eggs in 1× standard PBS, followed by sonication using Branson, B-12 sonifier probe, and centrifugation with 16,000 g at 4°C). The soluble supernatants were pooled and sterilized by a syringe filter (0.45 μm). The recombinant protein antigens were produced as previously described [26,41]. Briefly, expression plasmids of ShSerpin and RP26 were transformed into *Escherichia coli* BL21 (DE3), and proteins were expressed by isopropyl β-D-thiogalactopyranoside (IPTG) induction, followed by His-tag purification.

## Detection of IgG against S. haematobium antigens by ELISA

ELISA was conducted as previously described [36]. 96-well micro-well plates (Thermo Scientific, Roskilde, Denmark) were coated overnight at 4°C with 1 μg/mL of each antigen (ShSEA, ShSerpin, RP26, or cocktail antigen of ShSerpin and RP26; mixed in the mass ratio of 1:1). The plates were washed three times with PBS-T (0.05% Tween 20 in PBS, pH 7.6), then blocked with 150 μL of 1% casein-PBS for 2 h at room temperature. Then, 50 μL of standard reference sample dilution series and 1:1,000 diluted test plasma samples were applied to the plates and incubated for 1 h at 37°C. After washing with PBS-T, the plates were incubated with 50 μL of 1:10,000 diluted HRP-conjugated goat anti-human IgG (ab6858, Abcam plc, Cambridge, UK) for 1 h at 37°C. Subsequently, the plates were washed with PBS-T and incubated with 50 μL of 1-Step Ultra TMB-ELISA (Thermo Fisher Scientific, Rockford, USA) at room temperature for 20 min in the dark. Finally, 50 μL of 1 M sulfuric acid was added to stop the reaction. Absorbance was measured at 450 nm using a Multiskan FC microplate reader (Thermo Fisher Scientific, Vantaa, Finland).

To standardize the absorbance data obtained from every micro-well plate, an arbitrary unit (U) was defined using standard reference plasma as in the previous study [36]. In short, a

pooled mixture of 20 *S. haematobium*-infection positive plasma was set as a standard reference. The reference plasma was 3-fold serially diluted in nine steps starting from 1:1,000 and applied to all plates. An arbitrary concentration unit of specific IgG in the reference plasma was defined by calculating the dilution factor multiplied by 10,000 (10 units to 1:1,000 diluted reference plasma). Then standard curves based on 5-parameter logistic regression fits were generated using online software (www.elisaanalysis.com), and measured absorbance values of test samples were converted into arbitrary unit values by fitting the curves. All samples were tested in duplicate, and intra-assay variations between duplicates were calculated. The samples with % coefficient variation over 10% were re-analyzed. The cut-off units for samples being positive were set as the geometric mean plus 3 SD of non-endemic controls (n = 25, Japanese)' unit values. The cut-off values of the four antigen sets were; 0.459 U for ShSEA, 0.552 U for ShSerpin, 0.165 U for RP26 and 0.155 U for the ShSerpin-RP26 mixture.

## Samples for cross-reactivity evaluation

131 samples with other parasitic infections from *S. haematobium* non-endemic area were used to evaluate cross-reactivity. The 131 samples included samples included samples of other species of schistosome; schistosomiasis mansoni (n = 20), schistosomiasis japonica (n = 12), trematodes and cestodes; paragonimiasis (n = 20), fascioliasis (n = 20), clonorchiasis (n = 10), sparganosis (n = 19), and nematodes; gnathostomiasis (n = 10), and toxocariasis (n = 20). Schistosomiasis mansoni samples were from Mbita, a *S. mansoni* endemic area in Kenya [36], and non-schistosomiasis samples were collected in Japan. The samples positive for worms/ eggs were selected preferentially. Some schistosomiasis mansoni samples were negative for egg, but the samples were confirmed to be infected by the positive results of the point-of-care circulating cathodic antigen (POC-CCA) test as well as the positive results of antibody ELISA against *S. mansoni* SEA. A substantial number of schistosomiasis japonica, paragonimiasis, fascioliasis, clonorchiasis, gnathostomiasis and toxocariasis were without evidence of worms/ eggs. These cases were selected based on clinical symptoms and high antibody levels against the relevant parasites' antigens in ELISA.

## Data analysis

Data analyses and visualizations were carried out on GraphPad Prism 7.04 and R ver. 4.0.2 [42]. Mann-Whitney's U tests were used to compare the IgG levels between *S. haematobium* infected (Sh+) and the uninfected (Sh-) groups. To evaluate the diagnostic performances of the antigens, receiver-operating characteristic (ROC) analyses were applied to the 269 samples collected in Kwale (Sh+ and Sh-). Kruskal-Wallis tests were performed to compare antibody levels between different infection intensity groups. Spearman's rank correlation coefficient analyses were used to evaluate the correlation between the CAA concentrations and the antibody levels. Statistical significance was set at $p < 0.05$.

## Results

### The prevalence of S. haematobium infection among 269 pupils by urine egg and plasma CAA measurement

The prevalence and the infection intensity among 269 pupils determined by the egg detection in urine and the CAA measurement in plasma (PCAA) are shown in **Table 1**. The prevalence of *S. haematobium* infection determined by urine egg detection was 14.1% (38/269). The geometric mean of egg excretion among the egg-positive individuals was 9 (eggs/10 mL urine), and most of them (32/38) harbored light-intensity infection (1–49 eggs/10 mL urine,

according to WHO classification [4]). In contrast, PCAA500 was positive in 50.2% (135/269), much higher than the egg-positive proportion. The egg-positive individuals were also positive for PCAA500, except five pupils harboring light-intensity infection.

Nineteen individuals showed PCAA indecisive results and no egg excretion. As these cannot be decided as truly positive or negative, we analyzed the results under two conditions, regarding PCAA indecisive cases as either negative (Indec-) or positive (Indec+). Under Indec- condition, 141 (52.4%) were categorized as Sh+ and 128 (47.6%) were Sh-. Under Indec + condition, 160 (59.5%) were Sh+ and 109 (40.5%) were Sh-.

Kato-Katz slides were available from 262 individuals. Regarding STH infection, hookworm infections were the most prevalent (12/262, 4.5%). Of these 12 individuals, six were co-infected with *S. haematobium*. In addition, *Trichuris trichiura* infections were positive in three individuals (1.1%), and one had a co-infection with *S. haematobium* (S1 Table).

## Diagnostic performances of IgG detection against S. haematobium antigens (ShSEA, ShSerpin, RP26, and ShSerpin-RP26 mixture)

Next, we analyzed the IgG levels against ShSEA, ShSerpin, RP26 and ShSerpin-RP26 mixture in 269 plasma samples by ELISA. The cut-off units for samples being positive were set as the geometric mean plus 3 SD of non-endemic controls' unit values. The cut-off values of the four antigen sets were; 0.459 U for ShSEA, 0.552 U for ShSerpin, 0.165 U for RP26 and 0.155 U for the ShSerpin-RP26 mixture. The sensitivity and the specificity of IgG detection against each antigen were calculated based on the composite reference results (Sh+ or Sh-). Data plots of IgG levels to each antigen and their diagnostic performances under Indec- condition are presented in Fig 2 and Table 2. The results under Indec+ condition are shown in S1 Fig and S2 Table. The IgG levels were significantly higher in the *S. haematobium* infected group (Sh+) compared to the uninfected group (Sh-) for all antigens tested in this study (Figs 2 and S2). Regardless of active infection status (Sh+ or Sh-), anti-ShSEA IgG levels in most of Kwale samples were higher than the cut-off value and showed the highest sensitivity (99.3%, under Indec- condition unless otherwise stated) and the poorest specificity (16.4%) among the four antigen sets (Fig 2 and Table 2). The accuracy of anti-ShSEA IgG detection was therefore limited to 59.9%. The single recombinant antigens showed lesser sensitivity (66.7% for ShSerpin and 82.3% for RP26), but the specificities were far better than ShSEA, especially for ShSerpin (85.9%). The ShSerpin-RP26 mixture showed improved sensitivity of 88.7% and specificity of 67.2%, yielding the highest diagnostic accuracy of the four antigens tested in this study (78.4%, Table 2). From ROC analyses, the areas under the curve (AUC) of ShSEA and the ShSerpin-RP26 mixture were almost equal (0.884 and 0.887, respectively) (Fig 3). The sensitivities of the four antigens were lower under Indec+ than Indec- condition, because a considerable number of the CAA indecisive samples showed negative results for IgG detection (S3 Table). Even under Indec+ condition, the ShSerpin-RP26 mixture had the best diagnostic accuracy (79.9% accuracy: 76.9% sensitivity and 84.4% specificity) (S2 Table and S2 Fig).

## Correlation between the plasma CAA concentrations and the IgG levels against S. haematobium antigens

The CAA concentration and the number of worms in the host have reported to be well correlated [43], and we next examined the IgG levels' distributions in different categories of infection intensities determined by PCAA500 results (Fig 4). For all antigens, there were significant differences in antibody levels between each infection intensity group, and the group with higher CAA concentrations showed higher IgG levels against ShSEA, ShSerpin and the ShSerpin-RP26 mixture. The correlations between CAA concentrations and the IgG levels were also

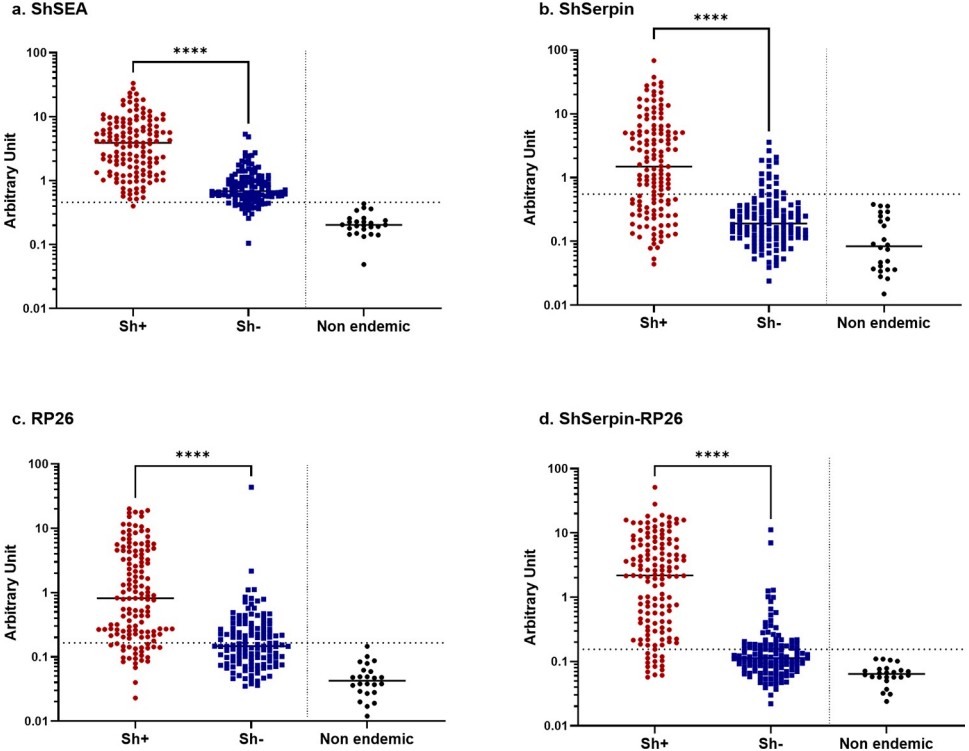

**Fig 2. Individuals with active *S. haematobium* infection showed higher IgG levels against the four antigen sets.**
Total IgG levels against (a) ShSEA, (b) ShSerpin, (c) RP26, (d) ShSerpin-RP26 mixture were analyzed among
individuals from Kwale with active *S. haematobium* infection (Sh+, n = 141, shown in red dots) and without active
infection (Sh-, n = 128, blue dots) determined by the composite reference (CAA Indecisive results regarded as CAA
negative, Indec-). The non-endemic control samples are included as the reference (Non endemic, n = 25, black dots).
The dotted lines show the cut-off values determined by the geometric mean plus 3 SD of non-endemic controls
(Japanese, n = 25)' unit values. The cut-off value of each antigen was; 0.459 for ShSEA, 0.552 for ShSerpin, 0.165 for
RP26, and 0.155 for ShSerpin-RP26 mixture. The antibody levels of the Sh+ and Sh- groups were compared by using
Mann-Whitney's U tests. Statistical significance was set at p < 0.05 and is shown using asterisks: **** = p < 0.0001.
The horizontal bars represent the median values of arbitrary units of each group. Sh+: *S. haematobium* infection
positive (pCAA500 positive and/or urine egg positive), shown in red dots (n = 141). Sh-: *S. haematobium* infection
negative (pCAA500 negative and urine egg negative), shown in blue dots (n = 128). NC: non-endemic controls
(Japanese), shown in black dots (n = 25).

analyzed, and there were statistically significant positive correlations between the CAA concentrations and the arbitrary units of IgG level against the four sets of antigens (ShSEA,
r = 0.729; ShSerpin, r = 0.630; RP26, r = 0.540; ShSerpin-RP26, r = 0.695. p < 0.0001. **Fig 5**).
ShSEA, ShSerpin and ShSerpin-RP26 mixture showed better correlations compared to RP26.

## Cross-reactivity with other helminthic infections

We tested the cross-reactivity of the four antigen sets with 131 samples positive for other helminthic infections; schistosomiasis mansoni, schistosomiasis japonica, paragonimiasis, fascioliasis, clonorchiasis, sparganosis, gnathostomiasis, and toxocariasis from *S. haematobium* nonendemic areas (**Table 3** and **Fig 6**). All the antigen sets showed more or less cross-reactivity
with other helminthic infections. ShSEA presented the highest cross-reactivity (41.2% of 131
samples and 34.3% of 99 non-schistosome-infected samples) as expected and showed the reaction with all the schistosomiasis mansoni samples. On the other hand, the recombinant antigens, ShSerpin and RP26, exhibited lower cross-reactivities, corroborating these antigens
showed higher specificities compared to ShSEA. ShSerpin showed the lowest cross-reactivities

**Table 2. Diagnostic performance of total IgG detection against ShSEA, ShSerpin, RP26 and ShSerpin-RP26 mix.**

| Antigen | | Indec- | | |
|---|---|---|---|---|
| | | Sh+* (n = 141) | Sh–[†] (n = 128) | PPV, NPV (95% CI) |
| ShSEA | Positive (n = 247) | 140 | 107 | PPV: 56.7% (54.8–58.6) |
| | Negative (n = 22) | 1 | 21 | NPV: 95.5% (74.1–99.4) |
| | Sn, Sp (95% CI) | Sn: 99.3% (96.1–100) | Sp: 16.4% (10.5–24.0) | Acc: 59.9% (53.7–65.8) |
| ShSerpin | Positive (n = 112) | 94 | 18 | PPV: 83.9% (77.0–89.1) |
| | Negative (n = 157) | 47 | 110 | NPV: 70.1% (64.7–74.9) |
| | Sn, Sp (95% CI) | Sn: 66.7% (58.2–74.4) | Sp: 85.9% (78.7–91.5) | Acc: 75.8% (70.3–80.8) |
| RP26 | Positive (n = 172) | 116 | 56 | PPV: 67.4% (62.3–71.9) |
| | Negative (n = 97) | 25 | 72 | NPV: 74.2% (66.2–80.9) |
| | Sn, Sp (95% CI) | Sn: 82.3% (75.0–88.2) | Sp: 56.3% (47.2–65.0) | Acc: 69.9% (64.0–75.3) |
| ShSerpin-RP26 mixture | Positive (n = 167) | 125 | 42 | PPV: 74.9% (69.8–79.3) |
| | Negative (n = 102) | 16 | 86 | NPV: 84.3% (76.9–89.7) |
| | Sn, Sp (95% CI) | Sn: 88.7% (82.2–93.4) | Sp: 67.2% (58.3–75.2) | Acc: 78.4% (73.0–83.2) |

The sensitivity and the specificity were calculated by using the cut-off values determined by geometric mean + 3 SD of non-endemic controls. CAA500 indecisive results were regarded as CAA negative (Indec-). Thus, PCAA500 indecisive and urine egg negative samples (n = 19) were not included in Sh+.

AUC, area under the curve; Sn, sensitivity; Sp, specificity; PPV, positive predictive value; NPV, negative predictive value; Acc, accuracy.

*Sh+: *S. haematobium* infection positive (PCAA500 positive and/or urine egg positive)

[†]Sh-: *S. haematobium* infection negative (PCAA500 negative and urine egg negative)

with the test samples (13.7% of the 131 samples and 9.1% of the 99 non-schistosome-infected samples), and notably, the reactions with schistosomiasis mansoni samples were limited. By mixing the two antigens, the cross-reactivities of ShSerpin-RP26 slightly increased, but the results were better than those of ShSEA. The samples that cross-reacted with the recombinant protein antigens (ShSerpin, RP26 and ShSerpin-RP26 mixture) also showed high antibody level against ShSEA. These results were similar to our previous study (36). Overall, a considerable number of the *S. mansoni* infection-positive samples showed reaction with tested antigens, while *S. japonicum* samples did not show such strong reactivities. Among the non-schistosomiasis samples, some specific samples of the trematode infections such as paragonimiasis, fascioliasis and clonorchiasis showed high antibody levels against four *S. haematobium* antigen sets. However, the cross-reactions with the nematode infection, such as gnathostomiasis and toxocariasis, were limited.

Then, we assessed STH infection-positive plasma samples with or without *S. haematobium* infections collected in Kwale. The sample size was limited (n = 14; 11 samples with hookworm infections and three samples with *T. trichiura* infections. One sample with CAA indecisive result was excluded. **S1 Table**). As shown in **S3 Fig**, most of the *S. haematobium* infection negative samples with STH infection (Sh-STH+) were positive for anti-ShSEA antibody. Still, it

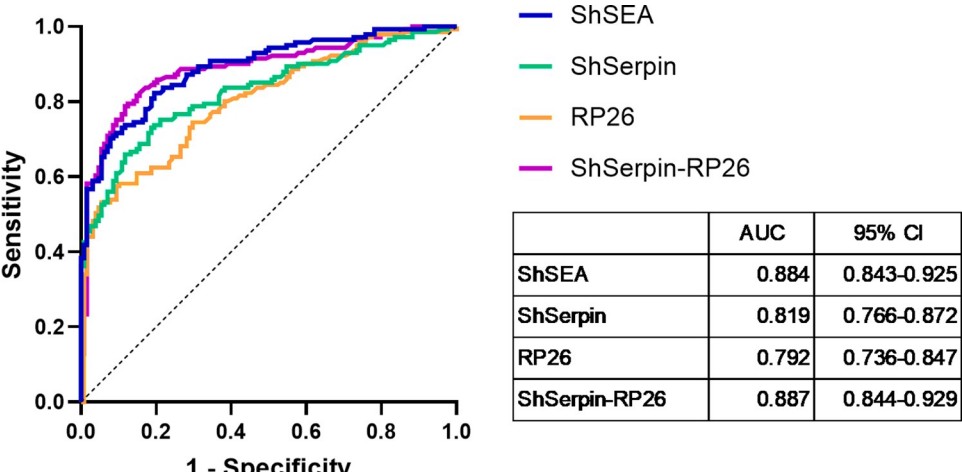

| | AUC | 95% CI |
|---|---|---|
| **ShSEA** | 0.884 | 0.843–0.925 |
| **ShSerpin** | 0.819 | 0.766–0.872 |
| **RP26** | 0.792 | 0.736–0.847 |
| **ShSerpin-RP26** | 0.887 | 0.844–0.929 |

**Fig 3. The ShSerpin-RP26 mixture capacity detecting active infections was similar to ShSEA.** The ROC curves were generated from the ELISA results of Kwale samples infected (Sh+, n = 141) and uninfected (Sh-, n = 128) with *S. haematobium*. pCAA500 indecisive results were considered as CAA negative (Inde-). The area under curve (AUC) of ShSEA, ShSerpin, RP26 and ShSerpin-RP26 mixture was 0.884, 0.819, 0.792 and 0.887, respectively.

cannot be determined if the results were due to cross-reaction with STH or past schistosome infection. The number of Sh-STH+ samples showing positive IgG to anti-recombinant schistosome protein antigens (ShSerpin, RP26, ShSerpin-RP26 mixture) was smaller than that of samples reactive with ShSEA, while some were positive indicating past schistosome infection.

## Discussion

In the present study, we evaluated antibody against the schistosome recombinant (ShSerpin and RP26) and crude (ShSEA) antigens in human plasma samples collected from *S. haematobium* endemic areas under intensified interventions. In the three schools investigated, the egg positive proportion dropped from 24.2% (51/211 in 2012) to 14.1% (38/269 in 2019, the present study) [37]. Thus, the prevalence and intensity of *S. haematobium* infection were considered to be declining at our study site. School children had been offered annual MDA since 2012 in schistosomiasis endemic areas in Kenya [3], and our observation of the reduced egg-positive rate may have reflected the successful outcome of the deworming program. To develop an infection detecting tool for monitoring transmission, we selected the population with declining infection intensity as our target. As it has been reported that the sensitivity of egg detection is significantly impaired under intensified interventions [10], we used CAA detection, together with egg detection, as a reference tests to determine active infection status. As shown in **Table 1**, over 80% of our study participants were negative for egg excretion, while half were positive for CAA in plasma.

The IgG to crude ShSEA and the mixture of recombinant ShSerpin and RP26 detected *S. haematobium* infections with high sensitivity (99.3% and 88.7% respectively, under the CAA Indec- condition, **Table 2**), which indicates that antibody detection against these antigens function even in low endemic settings. The sensitivity of the recombinant ShSerpin-RP26 mixture was lower than that of ShSEA (**Table 2**), while the detection capacities are suggested to be almost equal by the ROC curves (AUCs were 0.890 and 0.888, respectively). Moreover, the recombinant protein antigens showed fewer cross reactions to other helminthic infections and also are more suitable for mass production of qualified products than crude antigens such as SEA. Therefore, we suggest that the mixture of ShSerpin and RP26 may be useful as an

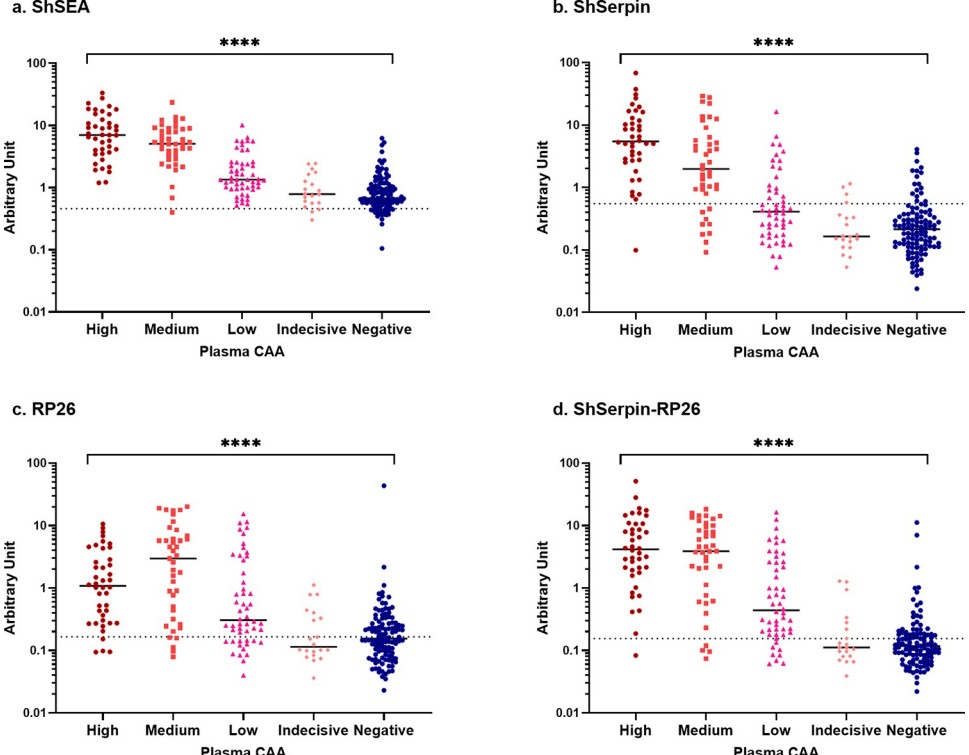

**Fig 4. IgG levels against *S. haematobium* antigens were associated with the infection intensity by plasma CAA.**
Total IgG levels against (a) ShSEA, (b) ShSerpin, (c) RP26, (d) ShSerpin-RP26 mixture were analyzed among Kwale samples with different categories of infection intensity determined by plasma CAA concentrations (pCAA500). CAA High: pCAA500 $\geq$ 100 pg/mL (n = 42); CAA Medium: 10 $\leq$ pCAA500 < 100 pg/mL (n = 41); CAA Low:1 $\leq$ pCAA500 < 10 pg/mL (n = 52); CAA Indecisive: 0.5 $\leq$ pCAA500 < 1 pg/mL (n = 20); CAA Negative: 0 $\leq$ pCAA500 < 0.5 pg/mL (n = 114). Kruskal-Wallis tests were performed for comparing antibody levels between different infection intensity groups measured by PCAA. Statistical significance was set at p < 0.05 and is shown using asterisks: **** = p < 0.0001. The dotted lines show the cut-off values determined by the geometric mean plus 3 SD of non-endemic controls' unit values. The cut-off value of each antigen was; 0.459 for ShSEA, 0.552 for ShSerpin, 0.165 for RP26, and 0.155 for ShSerpin-RP26 mixture. The horizontal bars represent the median values of arbitrary units of each group.

alternative to ShSEA to monitor *S. haematobium* transmission by detecting light-intensity infections in endemic areas. Antibody detection against either ShSerpin alone was reported to be highly sensitive to detect *S. haematobium* infection in the previous report [26]. In this study, we expected that the performance of ShSerpin would be hampered, because we targeted a population with light infection intensity and used highly sensitive tests as references. The sensitivity of ShSerpin (66.7% under the Indec- condition, **Table 2**) was lower than previous study by Tanigawa et al. [26] as expected. On the other hand, RP26 showed better sensitivity (82.3% under the Indec- condition, **Table 2**). When compared with our previous report on detecting *S. mansoni* infection in a low endemic setting [36], we noticed that RP26 demonstrated increased sensitivity to detect *S. haematobium* infection. This suggested that RP26 is more suitable to detect *S. haematobium* infection than *S. mansoni*, as shown in the study by Hancock et al. [32]. Alternatively, the increased sensitivity when applying RP26 for detection of *S. haematobium* infection may be due to the coincidental high numbers of samples with acute infection, as anti-RP26 IgG is reported to be produced in acute schistosome infection and then decrease in the chronic phase in the several studies [30,31]. The sensitivities of single recombinant ShSerpin and RP26 were limited, but the use of cocktail antigen improved

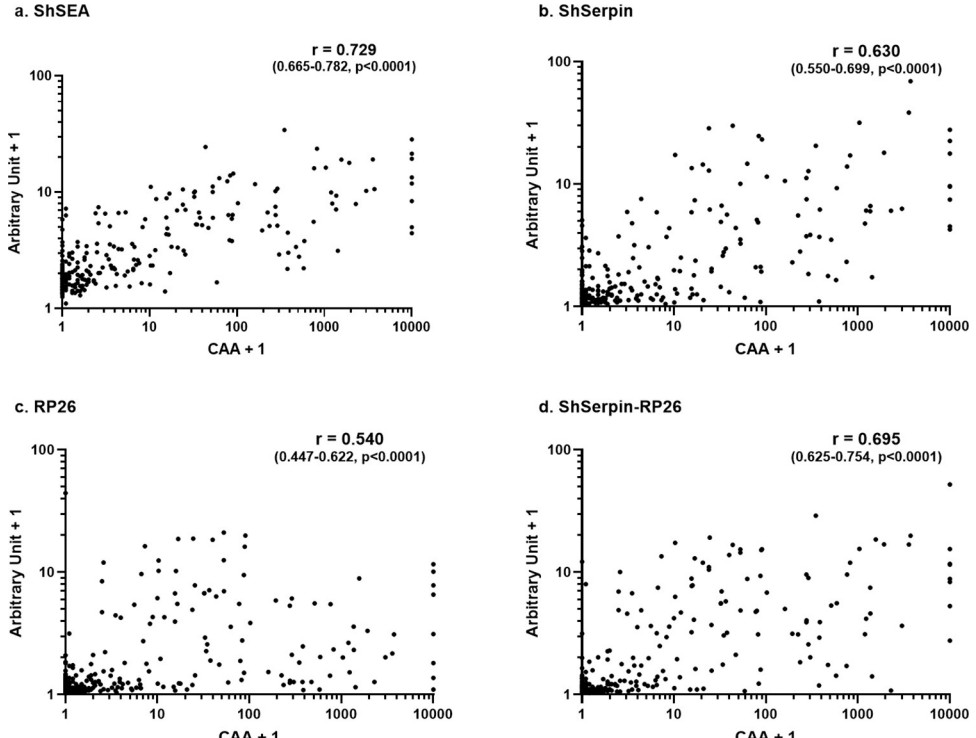

**Fig 5. Correlation between the CAA concentration by pCAA500 and the total IgG levels against ShSEA, ShSerpin, RP26 and ShSerpin-RP26 mixture.** The correlations between total IgG levels against (a) ShSEA, (b) ShSerpin, (c) RP26, (d) ShSerpin-RP26 mixture and CAA concentrations were analyzed using Spearman's rank correlation coefficient analyses. The correlation coefficient of ShSEA, ShSerpin, RP26 and ShSerpin-RP26 mixture was 0.729 (95% CI 0.665–0.782), 0.630 (95% CI 0.550–0.699), 0.540 (95% CI 0.447–0.622) and 0.695 (95% CI 0.625–0.754) respectively. For the all antigen sets, the p values were below 0.0001. As the axes are logarithmic, actual CAA concentration + 1 were plotted in the figures.

**Table 3. Number and proportion of cross-reactive samples from different helminthic infections.**

| Antigen | Schisto | | Non-Schisto | | | | | | Non-Schisto (%) | All (%) |
|---|---|---|---|---|---|---|---|---|---|---|
| | Sm (n = 20) | Sj (n = 12) | Pw (n = 20) | Fh (n = 20) | Cs (n = 10) | Se (n = 19) | Gd (n = 10) | Tc (n = 20) | | |
| **ShSEA** | 20 | 9 | 4 | 12 | 5 | 5 | 3 | 5 | 34 (34.3) | 54 (41.2) |
| **ShSerpin** | 5 | 4 | 3 | 3 | 2 | 0 | 1 | 0 | 9 (9.1) | 18 (13.7) |
| **RP26** | 14 | 2 | 3 | 3 | 3 | 0 | 1 | 0 | 10 (10.1) | 26 (19.8) |
| **ShSerpin-RP26 mixture** | 13 | 4 | 3 | 2 | 3 | 3 | 1 | 0 | 12 (12.1) | 29 (22.1) |

Total IgG levels against ShSEA, ShSerpin, RP26, ShSerpin-RP26 mix were analyzed among 131 samples with helminthic infections. The samples were from patients with schistosomiasis mansoni (Sm, n = 20), schistosomiasis japonica (Sj, n = 12), paragonimiasis (Pw, n = 20), fascioliasis (Fh, n = 20), clonorchiasis (Cs, n = 10), sparganosis (Se, n = 19), gnathostomiasis (Gd, n = 10), and toxocariasis (Tc, n = 20).

Number of samples which showed the IgG levels above the cut-off values are shown in the table. In "All" column, % shows the percentage of cross-reacted samples to the 131 test samples. "Non-schisto" column shows the number and percentage of non-schistosome samples (Pw, Fh, Cs, Se, Gd and Tc) cross-reacted. In this column, percentages to 99 non-schisto samples are shown.

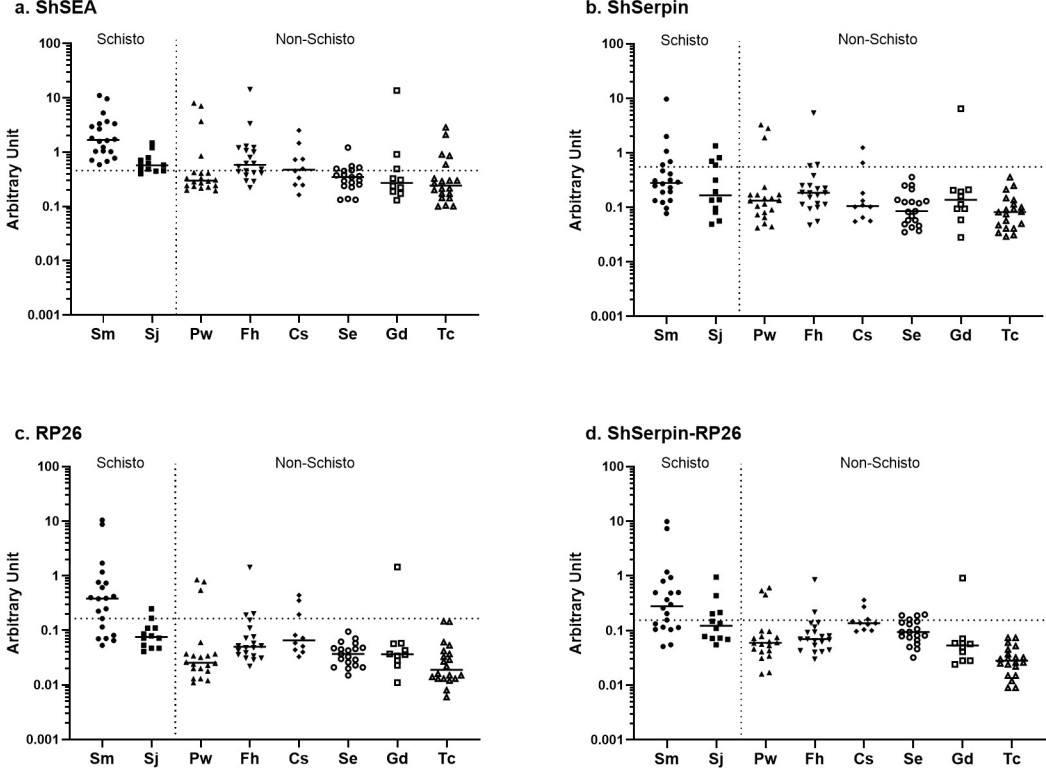

**Fig 6. Recombinant protein antigens (ShSerpin, RP26 and ShSerpin-RP26 mixture) had lower cross-reactivity compared to ShSEA.** Total IgG levels against (a) ShSEA, (b) ShSerpin, (c) RP26, (d) ShSerpin-RP26 mix were analyzed among 131 samples with helminthic infections. The samples were from patients with schistosomiasis mansoni (Sm, n = 20), schistosomiasis japonica (Sj, n = 12), paragonimiasis (Pw, n = 20), fascioliasis (Fh, n = 20), clonorchiasis (Cs, n = 10), sparganosis (Se, n = 19), gnathostomiasis (Gd, n = 10) and toxocariasis (Tc, n = 20). The dotted lines show the cut-off values determined by the geometric mean plus 3 SD of non-endemic controls' unit values. The cut-off value of each antigen was; 0.459 for ShSEA, 0.552 for ShSerpin, 0.165 for RP26, and 0.155 for ShSerpin-RP26 mixture. The horizontal bars represent the median values of arbitrary units of each type of parasite infection.

sensitivity, as reported in the previous studies [33–35]. The antigen sets, the number or ratio of mixed antigens, and the detection system should be optimized to develop a better test with higher sensitivity [33,44,45].

As antibodies against schistosome antigens remain at a high level even after the worm clearance and usually do not fully discriminate active from past infection [13]. To be used for monitoring real-time transmission dynamics, the effect of past infections is better to be minimized. Although we could not confirm the past infection cases among the participants in this study, many of them were suggested to have been exposed to *S. haematobium* considering their age and the endemicity of the region [37]. In this study, the majority of the individuals without active infection (Sh-) were positive for anti-ShSEA IgG (**Table 2 and Fig 2**). Egg trapped in host organs remain after PZQ treatment, and IgG to SEA usually stays elevated after treatment [46,47]. Therefore, the high anti-ShSEA IgG positivity among Sh- group may be partly resulted from past schistosome infections. On the other hand, the recombinant protein antigens showed better specificities (**Table 2**), which might indicate shorter longevity of the antibodies against ShSerpin or RP26 compared to that of SEA (and less cross-reactivity with other helminthic infections; discussed later). In the previous study using the murine model of *S. mansoni* infection, the IgG1 to recombinant *S. mansoni* Serpin and total IgG against RP26 was shown to decline in response to PZQ administration in murine model [47]. These observations

suggest the possibility of the rapid decline of anti-ShSerpin and anti-RP26 antibody following worm expulsion, which might lead to the better specificities compared to that of ShSEA. *Schistosoma* Serpins are detected in secreted/excreted products and tegument of all stages of schistosome [27,28], while the native protein of RP26 (Sm22.3) is expressed in cercariae, schistosomula, immature and adult worm, but not either in egg nor worm secretion/excretion products [30,31]. Our results suggests that worm expulsion by PZQ treatment may cause a rapid reduction of antigenic stimulation induced by worms. By mixing the two antigens, the specificity of ShSerpin-RP26 mixture was still far better than that of ShSEA, which indicates that ShSerpin-RP26 mixture is a good candidate for detecting active infections preferentially. For further clarifying the dynamics of the antibody to these schistosome antigens, studies analyzing antibody levels in human specimens before and after treatment need to be conducted.

We also examined the correlation between the plasma CAA concentration and the antibody levels against the *S. haematobium* antigens. Generally, the infection intensities cannot be assessed by the antibody levels [21,48]. In the present study, the antibody levels against the four *S. haematobium* antigens were associated with the infection intensity determined by the plasma CAA concentrations (**Fig 4**). We also observed positive correlations between the IgG levels against *S. haematobium* antigens (especially ShSEA, ShSerpin and ShSerpin-RP26 mixture) and the plasma CAA concentrations (**Fig 5**). Regarding ShSEA, it was surprising to observe the good correlation between anti-ShSEA IgG levels and infection intensities. Normally, due to previous infections, people in endemic areas show high antibody levels even after treatments [49]. At first, we thought that the infection negative (Sh-) participants from Kwale would show high anti-ShSEA antibody levels, and that there would be no correlation between anti-ShSEA IgG levels and infection intensities. The correlation observed in this study may be due to the decline of infection transmission in the study site, which is corroborated by the fact that some Sh- people showed no antibodies against ShSEA. Those participants might have stayed cured for years after successful treatment or have never been infected with schistosomes. Anti-ShSerpin IgG levels also showed a tendency to decline as the infection intensity decreases. It also might be partially because of the decline of anti-ShSerpin antibodies following worm expulsion besides the decline of schistosomiasis transmission in the study participants. In contrast, RP26, whose native protein is not expressed in the egg stage, presented a weaker correlation with infection intensities, suggesting that eggs induce stronger antibody responses compared to worms. Infection intensity is a crucial parameter of schistosomiasis morbidity [5,48], and our results suggest the antibody detection against ShSerpin (and ShSerpin-RP26 mixture) may also be useful in assessing the efficacy of morbidity and transmission control targeting the mass population.

Generally, *S. haematobium* endemic areas are also endemic for other helminthic diseases. Therefore, less cross-reactivity of the antibody test is desirable to reduce misdiagnosis and to assess the true endemicity of schistosomiasis. Schistosome SEA comprises multiple components, and its cross-reaction with other helminths has been reported [23,24,50,51]. As expected, ShSEA presented the highest cross-reactivity with other helminthic infections among the four antigens tested in this study (**Table 3** and **Fig 6**). We could only collect the *S. mansoni* and STH infection data based on the Kato-Katz technique and do not know other helminth infection status of our study population, but the poor specificity of the ShSEA in this study would partly be the result of cross-reactions among Sh- individuals. The recombinant protein antigens used in this study showed better specificities and lower cross-reactivities with other helminthic infections, while some trematode-infected samples showed false positive results. The reactivities with other *Schistosoma* species were increased by combining antigens. Especially in areas where both *S. haematobium* and *S. mansoni* are endemic, this antigen mixture needs to be used with caution, because it cannot discriminate the two species. However, the ShSerpin-RP26 mixture showed better result than that of ShSEA in terms of helminth

cross-reactivities, which makes it advantageous for this antigen mixture to be used in setting where various helminth infections are prevalent.

One ideal use case of this antigen mixture might be antibody detection among children. Recently, young children, before receiving school-based MDA, were revealed to be infected and seropositive for schistosomes [52,53]. Applying this tool to a very young population may demonstrate infection among the "neglected" age group. At the same time, it would help better understand the current transmission situation without being affected by the longevity of the antibody. By using recombinant proteins, mass production and quality control are feasible without maintaining schistosomes to generate SEA, and it would be a substantial benefit in providing monitoring tools for a large population. Lateral flow-based assays are applicable to antibody detection and might implement point-of-care kits using the ShSerpin-RP26 mixture.

In conclusion, our results showed that antibody detection using the mixture of two single recombinant antigens, ShSerpin and RP26, could be a functional *S. haematobium* infection transmission monitoring tool and an alternative to anti-ShSEA antibody detection with lower cross-reactivity. Owing to MDA, the infection intensity and prevalence of schistosomiasis are declining in endemic regions. Under such conditions, antibody detection using the ShSerpin-RP26 mixture would be helpful in the decision-making during the accomplishment of the control programs.

## Supporting information

**S1 Table. Other helminthic infection prevalence by Kato-Katz method.** Soil-transmitted helminth (hookworm, *Ascaris lumbricoides*, *Trichuris trichiura*) and *S. mansoni* infection among the 262 individuals whose K-K slide(s) (at least one slide) were available. The geometric mean EPGs among egg positive individuals was 46 EPGs for hookworm and 10 EPGs for *T. trichiura*. "Sh infection pos" means *S. haematobium* infection positive (pCAA500 positive and/or urine egg positive). "Sh infection indec" means pCAA500 indecisive and urine egg negative. "Sh infection neg" means *S. haematobium* infection negative (pCAA500 negative and urine egg negative).
(XLSX)

**S2 Table. Diagnostic performance of total IgG detection against ShSEA, ShSerpin, RP26 and ShSerpin-RP26 mix (Indec+).** The sensitivity and the specificity were calculated by using the cut-off values from ROC analyses. The cut-off values with maximum accuracies by ROC analyses were adopted. In this table, pCAA500 indecisive results were regarded as CAA positive, and pCAA500 indecisive and urine egg negative samples (n = 19) were included in Sh+. AUC, area under the curve; Sn, sensitivity; Sp, specificity; PPV, positive predictive value; NPV, negative predictive value; Acc, accuracy. *Sh+: *S. haematobium* infection positive (pCAA500 positive and/or urine egg positive) †Sh-: *S. haematobium* infection negative (pCAA500 negative and urine egg negative)
(XLSX)

**S3 Table. Results of antibody detection in pCAA500 indecisive and egg negative samples (n = 19).** The results of IgG detection against ShSEA, ShSerpin, RP26 and ShSerpin-RP26 mixture among PCAA indecisive and egg negative samples. The cut-off values were from ROC analysis (ROC) or the mean + 3SD of non-endemic controls (NC mean + 3SD). Pos, positive results; Neg; negative results.
(XLSX)

**S4 Table. Results of CAA detection in plasma samples, egg detection in stool by microscopy and IgG detection against ShSEA, ShSerpin, RP26 and ShSerpin-RP26 mix by ELISA.**

PCAA500, egg counts in stool and ELISA data of all of 269 indivisuals.
(XLSX)

**S1 Fig. Total IgG levels against four sets of antigens (ShSEA, ShSerpin, RP26, ShSerpin-RP26 mix) in 269 plasma samples from Kwale and 25 negative control samples (Indec+; CAA Indecisive results regarded as CAA positive).** Total IgG levels against (a) ShSEA, (b) ShSerpin, (c) RP26, (d) ShSerpin-RP26 mixture were analyzed among S. haematobium infected (Sh+)/uninfected (Sh-) Kwale samples and non-endemic control samples from Japan (NC). pCAA500 indecisive results are considered as pCAA positive. The bars represent the median values of arbitrary units in each group. The dotted lines show the cut-off values determined by the geometric mean plus 3 SD of non-endemic controls (Japanese, n = 25)' unit values. The cut-off value of each antigen was; 0.459 for ShSEA, 0.552 for ShSerpin, 0.165 for RP26, and 0.155 for ShSerpin-RP26 mixture. Statistical significance was set at $p < 0.05$ and is shown using asterisks: **** = $p < 0.0001$. Sh+, S. haematobium infection positive (pCAA500 positive and/or urine egg positive), shown in red dots (n = 160). Sh-, S. haematobium infection negative (pCAA500 negative and urine egg negative), shown in blue dots (n = 109). NC, non-endemic controls (Japanese), shown in black dots (n = 25).
(TIF)

**S2 Fig. ROC curve of each antigen under Indec+ (CAA Indecisive results regarded as CAA positive) condition.** The ROC curves were generated from the ELISA results of Kwale samples infected (Sh+, n = 160) and uninfected (Sh-, n = 109) with *S. haematobium*. pCAA500 indecisive results were considered as CAA positive. The area under curve (AUC) of ShSEA, ShSerpin, RP26 and ShSerpin-RP26 mixture was 0.850, 0.783, 0.760 and 0.845, respectively.
(TIF)

**S3 Fig. Total IgG levels against four sets of antigens (ShSEA, ShSerpin, RP26, ShSerpin-RP26 mix) in 14 plasma samples with STH infections from Kwale.** Total IgG levels against (a) ShSEA, (b) ShSerpin, (c) RP26, (d) ShSerpin-RP26 mixture were analyzed among STH infections positive plasma samples with or without *S. haematobium* infection from Kwale. The dotted lines show the cut-off values determined by the mean + 3SD of non-endemic controls' units. Sh+STH+, *S. haematobium* infection positive (PCAA500 positive and/or urine egg positive) and STH infection positive (confirmed by four Kato-Katz slides), shown in red dots (n = 7). Sh-STH-, *S. haematobium* infection negative (PCAA500 negative and urine egg negative) and STH infection negative (confirmed by four Kato-Katz slides), shown in blue dots (n = 7).
(TIF)

## Acknowledgments

We would like to express our deepest appreciation to all the participants in this study. Our gratitude also goes to the teachers and the parents/guardians who have shown understanding and supported our surveys. We are also grateful for the support and management by Kwale County Hospital and Education Office. Furthermore, this work was supported by the field surveyors/research assistants in Kwale and our laboratory members in the Department of Parasitology, NUITM and Department of Parasitology, LUMC. Finally, we show high appreciation to the director of KEMRI.

## Author Contributions

**Conceptualization:** Mio Kokubo-Tanaka, Shinjiro Hamano.

**Data curation:** Mio Kokubo-Tanaka, Evans Asena Chadeka, Benard Ngetich Cheruiyot.

**Formal analysis:** Mio Kokubo-Tanaka, Anna Overgaard Kildemoes, Claudia J. de Dood.

**Funding acquisition:** Sammy M. Njenga, Remco de Vrueh, Cornelis Hendrik Hokke, Shinjiro Hamano.

**Investigation:** Mio Kokubo-Tanaka, Anna Overgaard Kildemoes, Evans Asena Chadeka, Benard Ngetich Cheruiyot, Miho Sassa, Claudia J. de Dood.

**Methodology:** Mio Kokubo-Tanaka, Taeko Moriyasu, Risa Nakamura, Mihoko Kikuchi, Yoshito Fujii, Claudia J. de Dood, Paul L. A. M. Corstjens.

**Project administration:** Shinjiro Hamano.

**Resources:** Risa Nakamura, Mihoko Kikuchi, Yoshito Fujii, Claudia J. de Dood, Paul L. A. M. Corstjens, Satoshi Kaneko, Haruhiko Maruyama, Sammy M. Njenga, Shinjiro Hamano.

**Supervision:** Cornelis Hendrik Hokke, Shinjiro Hamano.

**Validation:** Mio Kokubo-Tanaka.

**Visualization:** Mio Kokubo-Tanaka.

**Writing – original draft:** Mio Kokubo-Tanaka, Anna Overgaard Kildemoes, Cornelis Hendrik Hokke, Shinjiro Hamano.

**Writing – review & editing:** Mio Kokubo-Tanaka, Anna Overgaard Kildemoes, Paul L. A. M. Corstjens, Cornelis Hendrik Hokke, Shinjiro Hamano.

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
