## [Decision Letter · Decision Letter 0]

11 Nov 2024

PNTD-D-24-01409Detection of Serpin and RP26 specific antibodies is a promising approach for monitoring Schistosoma haematobium transmissionPLOS Neglected Tropical Diseases Dear Dr. Hamano, Thank you for submitting your manuscript to PLOS Neglected Tropical Diseases. After careful consideration, we feel that it has merit but does not fully meet PLOS Neglected Tropical Diseases's publication criteria as it currently stands. Therefore, we invite you to submit a revised version of the manuscript that addresses the points raised during the review process. Please submit your revised manuscript within 30 days Jan 10 2025 11:59PM. If you will need more time than this to complete your revisions, please reply to this message or contact the journal office at plosntds@plos.org. Please include the following items when submitting your revised manuscript:*
A rebuttal letter that responds to each point raised by the editor and reviewer(s). You should upload this letter as a separate file labeled 'Response to Reviewers'. This file does not need to include responses to any formatting updates and technical items listed in the 'Journal Requirements' section below.*
A marked-up copy of your manuscript that highlights changes made to the original version. You should upload this as a separate file labeled 'Revised Manuscript with Track Changes'.*
An unmarked version of your revised paper without tracked changes. You should upload this as a separate file labeled 'Manuscript'. If you would like to make changes to your financial disclosure, competing interests statement, or data availability statement, please make these updates within the submission form at the time of resubmission. Guidelines for resubmitting your figure files are available below the reviewer comments at the end of this letter. We look forward to receiving your revised manuscript. Kind regards, Maria Angeles Gómez-Morales, PhDAcademic EditorPLOS Neglected Tropical Diseases Eva ClarkSection EditorPLOS Neglected Tropical Diseases

Shaden Kamhawi

co-Editor-in-Chief

Paul Brindley

co-Editor-in-Chief

 **Journal Requirements:** **Additional Editor Comments (if provided):****Reviewers' comments:** Reviewer's Responses to Questions

**Key Review Criteria Required for Acceptance?**

**Methods**

-Are the objectives of the study clearly articulated with a clear testable hypothesis stated?

-Is the study design appropriate to address the stated objectives?

-Is the population clearly described and appropriate for the hypothesis being tested?

-Is the sample size sufficient to ensure adequate power to address the hypothesis being tested?

-Were correct statistical analysis used to support conclusions?

-Are there concerns about ethical or regulatory requirements being met?

Reviewer #1: The paper is well written, the experiments are carefully performed and the conclusion is valid.

Reviewer #2: The method is clearly stated but I think it's good to have the binding affinity, limit of detection data to validate the performance of the biosensor. I am not sure of the recognition element and antigen. There should be a graphical representation of the abstract.

Reviewer #3: Tanaka et al seek a method to improve the specificity and sensitivity of S.haematobium active infection. This manuscript follows the group's 2021 paper investigating mansoni infection.

The authors perform a robust analysis of stool, plasma and urine samples in a school-aged cohort (age not reported?) to assess the utility of available recombinant Schistosoma antigens and lab-derived undefined antigens (soluble egg antigen) in definitive diagnosis of active S.h. infection.

It appears the authors adhered to ethical standards in enrolling juveniles for their study; study design and objectives are clearly defined.

**Results**

-Does the analysis presented match the analysis plan?

-Are the results clearly and completely presented?

-Are the figures (Tables, Images) of sufficient quality for clarity?

Reviewer #1: The paper is well written, the experiments are carefully performed and the conclusion is valid.

Reviewer #2: There is clarity in the presentation of results but binding affinity data will do more Justice to performance of the biosensor a

Reviewer #3: Questions for the authors:

For ELISAs, how does the inclusion of the (+) plasma reference in the assay aid in your interpretation? Can the non-endemic control alone be used for analysis?

Figure 5: Will it be helpful to know the value of the cutoff for each of the 4 assays?

Methods: what is the source of haematobium eggs to generate 'ShSEA'

Did the authors test different concentrations of the ELISA capture (1ug/mL); the ShSEA is an imperfect reagent, but will increasing capture conc. improve delineation for diagnosis of active infection?

**Conclusions**

-Are the conclusions supported by the data presented?

-Are the limitations of analysis clearly described?

-Do the authors discuss how these data can be helpful to advance our understanding of the topic under study?

-Is public health relevance addressed?

Reviewer #1: The paper is well written, the experiments are carefully performed and the conclusion is valid.

Reviewer #2: Well written.

Reviewer #3: In conclusion, the limits of the assay are discussed, but the data is relevant to our field given the limitations of detection systems. A more conservative title may capture the conclusions more aptly, such as: 'Detection and analysis of Serpin and RP26 IgG for monitoring Schistosoma haematobium transmission'

**Editorial and Data Presentation Modifications?**

Reviewer #1: The paper is well written, the experiments are carefully performed and the conclusion is valid. I will recommend "Minor Revision".

Reviewer #2: Minor

Reviewer #3: Minor edit in the References: #13 includes doi;

#14 includes doi; has word duplication

**Summary and General Comments**

Reviewer #1: Comments for author:

PNTD-D-24-01409

" Detection of Serpin and RP26 specific antibodies is a promising 1 approach for monitoring Schistosoma haematobium transmission " by Tanaka M. et al.

This paper describes a comparative analysis of four different antigen ELISA formats, soluble egg antigen (ShSEA), ShSerpin, RP26 and the mixture, in the detection of active infection with S. haematobium in school children in the disease-endemic areas of the Lake Victoria region of Kenya. Based on the results, the authors concluded that the ShSerpin-RP26 mixture had better sensitivity for detecting active infection and could potentially replace ShSEA for transmission monitoring in near-elimination settings. The paper is well written, the experiments are carefully performed and the conclusion is valid. I have just a few minor points that should be addressed in your paper:

ELISAs using the ShSerpin-RP26 mixture as antigen cross-react with sera from patients infected with S. mansoni. Therefore, it should be mentioned in the discussion that caution should be exercised when using this ELISA in areas where schistosomiasis mansoni and schistosomiasis bilharzia are prevalent.

The ability to detect active infection is an important feature of the ELISA using the ShSerpin-RP26 mixture as antigen. Figure S-6 is a key element in demonstrating this and should be shown in the text rather than in the Supplement.

Reviewer #2: The study is a significant contribution to biosensor development towards elimination of schistosomiasis

Reviewer #3: Thank you for the opportunity to review a well-constructed and robust manuscript.

PLOS authors have the option to publish the peer review history of their article (what does this mean?). If published, this will include your full peer review and any attached files.

Reviewer #1: No

Reviewer #2: No

Reviewer #3: No

---

## [Editor Report · Decision Letter 1]

27 Dec 2024

Dear Prof. Hamano,

We are pleased to inform you that your manuscript 'Detection and analysis of Serpin and RP26 specific antibodies for monitoring Schistosoma haematobium transmission' has been provisionally accepted for publication in PLOS Neglected Tropical Diseases.

Best regards,

Jong-Yil Chai

Section Editor

Eva Clark

Section Editor

Shaden Kamhawi

co-Editor-in-Chief

Paul Brindley

co-Editor-in-Chief

All points raised by the reviewers have been well addressed in this revised manuscript and now it is acceptable for publication.

---

## [Editor Report · Acceptance letter]

7 Jan 2025

Dear Prof. Hamano,

We are delighted to inform you that your manuscript, "Detection and analysis of Serpin and RP26 specific antibodies for monitoring Schistosoma haematobium transmission," has been formally accepted for publication in PLOS Neglected Tropical Diseases.

Best regards,

Shaden Kamhawi

co-Editor-in-Chief

Paul Brindley

co-Editor-in-Chief
